# Kinematically Constrained Jerk–Continuous S-Curve Trajectory Planning in Joint Space for Industrial Robots

**Guanglei Wu** [1,*] and **Ning Zhang** [2]

1   School of Mechanical Engineering, Dalian University of Technology, Dalian 116024, China
2   AVIC Optronics, No. 25, Kaiyuan West Road, Xigong District, Luoyang 471000, China
*   Correspondence: gwu@dlut.edu.cn

**Abstract:** This work deals with jerk–continuous trajectory planning for robotic manipulators by means of the fourth-order S-curve to ensure motion smoothness. The algorithm presented in this work can cause the acceleration and jerk to stay in a saturated state in order to improve the efficiency of a robot's programming and operation. Moreover, a multi-axis synchronization planning algorithm is proposed and integrated for enhanced motion stability in terms of generated synchronized and continuous motion trajectories, for which the effectiveness of the proposed trajectory planning algorithm is verified in both the joint and Cartesian spaces. The proposed algorithm does not involve any optimization procedures or iterative processes, as the kinematically constrained trajectory is generated by polynomial equations to realize the real-time motion control of robots. Moreover, the presented algorithm can generate the jerk continuity trajectory, rather than only the acceleration continuity, as in most reported works.

**Keywords:** trajectory planning; kinematic constraints; S-curve; jerk continuity; time synchronization





## 1. Introduction

With the development of automation technology and computer science, industrial robots have been increasingly used in automatic assembly, spot welding, palletizing, drilling, and material handling due to their outstanding flexibility and adaptability [1,2]. As an essential issue in robotic technology, trajectory planning plays an important role in the process of controlling a robot to accomplish required tasks, as the selection of the trajectory significantly affects the stability and reliability of the robot. In trajectory planning, full consideration of the kinematic constraints of a robot can reduce the wear of the actuator and improve the velocity and tracking accuracy. Therefore, kinematic constraints, such as the velocity, acceleration, and jerk of the robot, are commonly considered in trajectory planning [3]. Moreover, real-time motion control is essential for industrial robots; thus, trajectory planning with lower computational complexity and time consumption is preferred.

In order to realize the smooth and fast motion of a robot in the joint or task space, many research works have been reported, in which trajectory planning methods based on polynomial and spline curves are the most commonly adopted [4–8]. Lin et al. [9] presented a robot's cubic polynomial joint trajectory based on kinematic constraints, which could generate the shortest motion trajectory. One limitation of the previous method was that it could not guarantee the continuity of acceleration. Liu et al. [10] developed quintic polynomial trajectory planning in joint space, and the results showed that the quintic polynomial could continuously ensure smooth acceleration, but with a cusp in the jerk profile. In view of the foregoing issue, Tang et al. [11] divided the task trajectory into multiple segments and used a 4-4-7-4 polynomial (a 4-4-7-4 polynomial trajectory means that three via-points between the starting and end positions are selected to construct four segments of the total path, where the third segment adopts a seventh-order polynomial

equation to mathematically depict the trajectory, while the remaining segments adopt a fourth-order polynomial) to plan the trajectory by means of the sequential quadratic programming algorithm (SQP) in order to minimize the execution time, which could ensure the continuity and controllability of the joint jerk curve, together with a reduction in shock and residual vibration. Aiming at the high performance of robots in motion planning, trajectory planning algorithms have also been reported in combination with intelligent optimization algorithms. Machmudah et al. [12] adopted a genetic algorithm (GA) and particle swarm optimization (PSO) to search the feasible sixth-order polynomial trajectories in joint space in a complex geometric environment to optimize the time taken for motion planning while subject to dynamic constraints. Huang et al. [13] proposed a method of using quintic B-spline interpolation in the joint space, and the non-dominated sorting genetic algorithm (NSGA-II) was deployed to obtain the optimal time–jerk trajectory. Kucuk [14] integrated a cubic spline with a seven-order polynomial to construct a smooth motion trajectory with a minimal execution time and an initial zero-impact for serial and parallel robots by means of PSO. On the other hand, most of these trajectory planning methods, which were based on polynomials and splines, required the application of various numerical optimization techniques in order to find the optimal solutions, resulting in increased processing time and hardware resources, which heavily affected the real-time operation of the robots in their applications.

In light of the previous works, this work presents a smooth trajectory generation algorithm that uses a quartic polynomial to represent the motion trajectories of a robot's joints to realize the real-time motion control of the robot. Moreover, a multi-axis synchronization planning algorithm is integrated to ensure the motion stability. The major advantage of the proposed algorithm lies in the buffeting suppression for the robot joints with the generation of a smooth-motion trajectory (i.e., the $C_3$-continuity level, namely, jerk continuity)—which is subject to the optimal execution time without the use of an iterative optimization algorithm—for real-time control.

## 2. Jerk–Continuous Trajectory Planning

The design of trajectories is a central issue in motion planning for robotic applications—for instance, as depicted in Figure 1, a trajectory can be generated for the accomplishment of the task of a robot from the starting position $\mathbf{p}_s$ to ending position $\mathbf{p}_e$ in the Cartesian space, where $\mathbf{p}_s$ and $\mathbf{p}_e$ are the functions of the joint variables $\mathbf{q}_s = [\theta_1^s, \theta_2^s, \ldots]$ and $\mathbf{q}_e = [\theta_1^e, \theta_2^e, \ldots]$, respectively. Particularly for the point-to-point (PTP) task, the S-curve trajectory is a relatively better candidate for minimizing the residual vibrations during a robot's motions [15–17]. In this procedure, different robot performances will be shown, and they heavily depend on the different motion profiles of the actuated joints, which are solved by using the inverse kinematics.

In general, for a given set of constraints (i.e., the maximum values of motion parameters of the joints, e.g., velocity, acceleration, jerk, etc.), increased smoothness of the trajectory will lead to a higher trajectory generation time and computational complexity [18]. It is noteworthy that the total planning time can be reduced to the minimum if the jerk, acceleration, and velocity can reach the peak values rapidly and can stay constant as long as possible [6]; they are constrained to the limitations of the motion parameters. Therefore, the trajectory with the minimum time should include a segment with a constant velocity, acceleration, and jerk, for which the velocity, acceleration, and jerk can reach their allowable maximums. Henceforth, this section presents a modified S-curve trajectory planning algorithm with a continuous jerk for multi-degree-of-freedom (dof) industrial robots in order to obtain the optimal trajectory in terms of execution time and to ensure the motion smoothness while being subject to kinematic constraints.

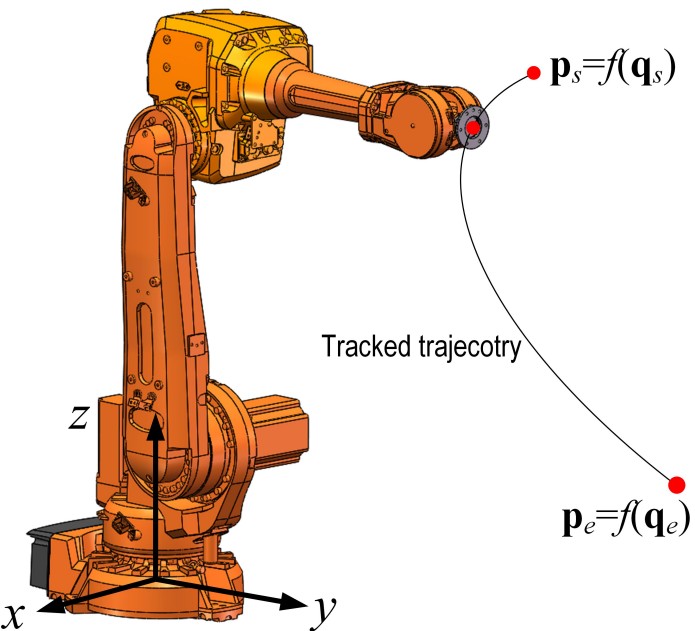

**Figure 1.** Tracked spatial trajectory of a robot end-effector from the starting position $\mathbf{p}_s$ to the ending position $\mathbf{p}_e$ (the original robot CAD model of IRB 4600 was downloaded from the official ABB website).

### 2.1. From Third-Order to Fourth-Order S-Curve Trajectories

The third-order S-curve trajectory was initially proposed by Castin and Paul [19], and it has become one of the most extensively used trajectory models due to the moderate complexity of the generation of the trajectory in the shortest planning time with the consideration of the jerk constraints.

As shown in Figure 2, the standard form of the third-order S-curve trajectory consists of seven segments, of which the first $[t_0, t_3]$ and the last three $[t_4, t_7]$ segments represent the acceleration and deceleration phases, respectively. The motion curves are produced by modifying the trapezoidal velocity profile by limiting the rising time of the acceleration profile [20]. According to the well-known Pontryagin maximum principle [21], the time-optimal jerk curve can be written in a Bang–Bang format, namely,

$$j(t) = \begin{cases} j_m & t_0 \leq t < t_1,\, t_6 \leq t < t_7 \\ 0 & t_1 \leq t < t_2,\, t_3 \leq t < t_4,\, t_5 \leq t < t_6 \\ -j_m & t_2 \leq t < t_3,\, t_4 \leq t < t_5 \end{cases} \tag{1}$$

where $j_m$ stands for the maximum jerk constraint, and the time-varying acceleration, velocity, and displacement of the trajectory can be calculated from the integral of the polynomial equation. Compared to the trajectory of a trapezoidal profile featuring a sharp acceleration, the rectangular jerk curve adopted with this motion trajectory can reduce the reaction forces on the robot joints caused by sudden acceleration, leading to a longer service life of the robot. On the other hand, this trajectory has the drawback of a jerk cusp, which will cause the robot to shake at the starting and ending positions of its movements, thus affecting the tracking accuracy. For applications requiring high precision, it is essential to ensure that the jerk curve is continuous to eliminate this shaking.

To handle this problem, a fourth-order S-curve trajectory planning model is introduced to divide each acceleration segment of the previous third-order S-curve trajectory into three independent stages, which causes the overall trajectory curve to have fifteen segments. The advantage of this algorithm is that a smooth trajectory can be planned while being subject to jerk continuity, and the optimal execution time can be solved according to the prescribed boundary conditions and kinematically constrained velocity, acceleration,

and jerk. Moreover, by making use of this algorithm, it is possible to realize real-time robot motion control, as it does not involve any iterative processes. Figure 3 depicts the normalized kinematic trajectory. Assuming that the maximum velocity, acceleration, and jerk are set to $v_{\max}$, $a_{\max}$, and $j_{\max}$, that the starting point and velocity are set to $q_s$ and $v_s$, and that the position and velocity at the ending position are denoted by $q_e$ and $v_e$, consequently, the jerk function of the trajectory can be defined as follows:

$$j(t) = \begin{cases} \frac{\tau_i}{T_i} j_{\max} & t_0 \leq t < t_1, \; t_{12} \leq t < t_{13} \\ j_{\max} & t_1 \leq t < t_2, \; t_{13} \leq t < t_{14} \\ \left(1 - \frac{\tau_i}{T_i}\right) j_{\max} & t_2 \leq t < t_3, \; t_{14} \leq t < t_{15} \\ 0 & t_3 \leq t < t_4, \; t_7 \leq t < t_8, \; t_{11} \leq t < t_{12} \\ -\frac{\tau_i}{T_i} j_{\max} & t_4 \leq t < t_5, \; t_8 \leq t < t_9 \\ -j_{\max} & t_5 \leq t < t_6, \; t_9 \leq t < t_{10} \\ -\left(1 - \frac{\tau_i}{T_i}\right) j_{\max} & t_6 \leq t < t_7, \; t_{10} \leq t < t_{11} \end{cases} \tag{2}$$

where $t_i$ ($i = 1, 2, \ldots, 15$) represents the instantaneous time of each movement transition, and $\tau_i = t - t_{i-1}$ represents the zero time at the transition point of each movement phase. The whole trajectory consists of fifteen segments, and the execution time of each segment is defined by $T_i = t_i - t_{i-1}$. The relationships among the jerk $j(t)$, acceleration $a(t)$, velocity $v(t)$, and displacement $q(t)$ of the fourth-order S-curve trajectory are given by

$$j(t) = \begin{cases} a(t) = a(t_i) + \int_{t_i}^{t} j(\tau_i) \mathrm{d}t \\ v(t) = v(t_i) + \int_{t_i}^{t} a(\tau_i) \mathrm{d}t \\ q(t) = q(t_i) + \int_{t_i}^{t} v(\tau_i) \mathrm{d}t \end{cases} \tag{3}$$

The parameterized motion functions with different time segments are expressed in Table 1.

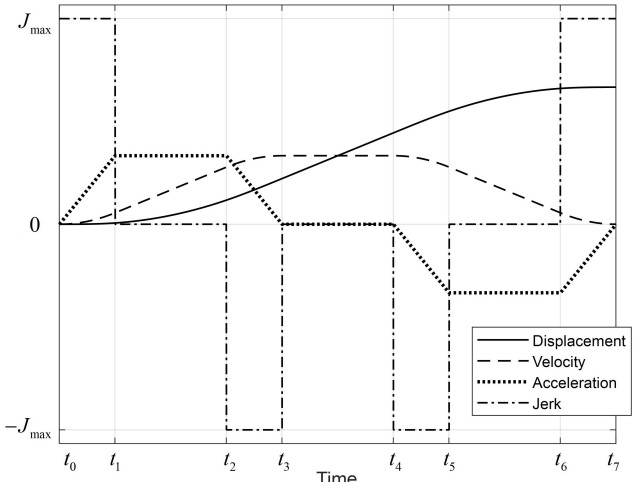

**Figure 2.** Kinematic trajectory of the third-order S-curve.

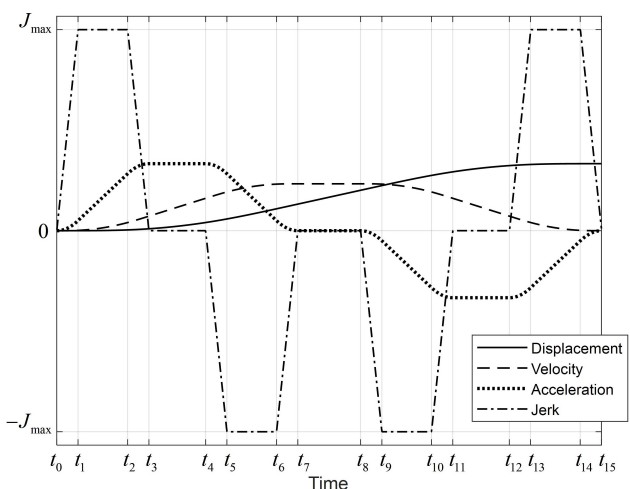

**Figure 3.** Kinematic trajectory of the fourth-order S-curve.

**Table 1.** The expression of the motion profiles with different time intervals and curve segments.

| Time | Function of Motion Profiles | Notations |
|---|---|---|
| $t \in [t_0, t_1]$ | $j(t) = \frac{j_{max}}{T_1}\tau_1$ <br> $a(t) = \frac{j_{max}}{2T_1}\tau_1^2$ <br> $v(t) = \frac{j_{max}}{6T_1}\tau_1^3 + v_s$ <br> $q(t) = \frac{j_{max}}{24T_1}\tau_1^4 + v_s\tau_1 + q_s$ | |
| $t \in [t_1, t_2]$ | $j(t) = j_{max}$ <br> $a(t) = j_{max}\tau_2 + \frac{j_{max}}{2}T_1$ <br> $v(t) = \frac{j_{max}}{2}\tau_2^2 + \frac{j_{max}}{2}T_1\tau_2 + v_1$ <br> $q(t) = \frac{j_{max}}{6}\tau_2^3 + \frac{j_{max}}{4}T_1\tau_2^2 + v_1\tau_2 + q_1$ | $v_1 = v_0 + \frac{j_{max}}{6}T_1^2$ <br> $q_1 = v_0 T_1 + \frac{j_{max}}{24}T_1^3$ |
| $t \in [t_2, t_3]$ | $j(t) = j_{max} - \frac{j_{max}}{T_3}\tau_3$ <br> $a(t) = j_{max}\tau_3 - \frac{j_{max}}{2T_3}\tau_3^2 + \frac{j_{max}}{2}(T_1 + 2T_2)$ <br> $v(t) = \frac{j_{max}}{2}\tau_3^2 - \frac{j_{max}}{6T_3}\tau_3^3 + \frac{j_{max}}{2}(T_1 + 2T_2)\tau_3 + v_2$ <br> $q(t) = \frac{j_{max}}{6}\tau_3^3 - \frac{j_{max}}{6T3}\tau_3^4 + \frac{j_{max}}{4}(T_1 + 2T_2)\tau_3^2 + v_2\tau_3 + q_2$ | $v_2 = v_1 + \frac{j_{max}}{2}T_2(T_1 + T_2)$ <br> $q_2 = q_1 + \frac{j_{max}}{12}T_2^2(2T_2 + 3T_1) + v_1T_2$ |
| $t \in [t_3, t_4]$ | $j(t) = 0$ <br> $a(t) = a_{max}$ <br> $v(t) = a_{max}\tau_4 - v_3$ <br> $q(t) = \frac{a_{max}}{2}\tau_4^2 + v_3\tau_4 + q_3$ | $v_3 = v_2 + \frac{j_{max}}{6}T_3(3T_1 + 6T_2 + 2T_3)$ <br> $q_3 = q_2 + \frac{j_{max}}{8}T_3^2(2T_1 + 4T_2 + T_3) + v_2T_3$ |
| $t \in [t_4, t_5]$ | $j(t) = -\frac{j_{max}}{T_5}\tau_5$ <br> $a(t) = -\frac{j_{max}}{2T_5}\tau_5^2 + a_{max}$ <br> $v(t) = -\frac{j_{max}}{6T_5}\tau_5^3 + a_{max}\tau_5 + v_4$ <br> $q(t) = -\frac{j_{max}}{24T_5}\tau_5^4 + \frac{a_{max}}{2}\tau_5^2 + q_4$ | $v_4 = v_3 + a_{max}T_4$ <br> $q_4 = q_3 + \frac{a_{max}}{2}T_4^2 + v_3T_4$ |
| $t \in [t_5, t_6]$ | $j(t) = -j_{max}$ <br> $a(t) = -j_{max}\tau_6 + a_{max} - \frac{j_{max}}{2}T_5$ <br> $v(t) = -\frac{j_{max}}{2}\tau_6^2 + \left(a_{max} - \frac{j_{max}}{2}\tau_5\right)\tau_6 + v_5$ <br> $q(t) = -\frac{j_{max}}{6}\tau_6^3 + \frac{1}{2}\left(a_{max} - \frac{j_{max}}{2}\tau_5\right)\tau_6^2 + v_5\tau_6 + q_5$ | $v_5 = v_4 - \frac{a_{max}}{6}T_5^2 + a_{max}T_5$ <br> $q_5 = q_4 - \frac{j_{max}}{24}T_5^3 + \frac{a_{max}}{2}T_5^2 + v_4T_5$ |
| $t \in [t_6, t_7]$ | $j(t) = -j_{max} + \frac{j_{max}}{T_7}\tau_7$ <br> $a(t) = -j_{max}\tau_7 + \frac{j_{max}}{2T_7}\tau_7^2 + a_{max} - \frac{j_{max}}{2}T_5 - j_{max}T_6$ <br> $v(t) = -\frac{j_{max}}{2}\tau_7^2 + \frac{j_{max}}{6T_7}\tau_7^3 + \left(a_{max} - \frac{j_{max}}{2}T_5 - j_{max}T_7\right)\tau_7 + v_6$ <br> $q(t) = -\frac{j_{max}}{6}\tau_7^3 - \frac{j_{max}}{24T_7}\tau_7^4 + \frac{2a_{max} - j_{max}T_5 - 2j_{max}T_1}{4}\tau_7^2 + v_6\tau_7 + q_6$ | $v_6 = v_5 - \frac{j_{max}}{2}T_6^2 + \left(a_{max} - \frac{j_{max}}{2}T_5\right)T_6$ <br> $q_6 = q_5 + \frac{6a_{max} - 3j_{max}T_5 - 2j_{max}T_6}{12}T_6^2 + v_5T_6$ |
| $t \in [t_7, t_8]$ | $j(t) = 0$ <br> $a(t) = 0$ <br> $v(t) = v_7$ <br> $q(t) = v_7\tau_8 + q_7$ | $v_7 = v_6 - \frac{j_{max}}{3}T_7^2 + \left(a_{max} - \frac{j_{max}}{2}T_5 - j_{max}T_6\right)T_7$ <br> $q_7 = q_6 - \frac{j_{max}}{8}T_7^3 + \frac{2a_{max} - j_{max}T_5 - 2j_{max}T_6}{4}T_7^2$ |

The motion profiles of the deceleration phase are symmetrical to those of the acceleration phase, which can also be obtained with an integration operation, which will not be described in detail. It can be seen in Figure 3 that in the acceleration phase, the equations of time intervals $T_1 = T_3 = T_5 = T_7$ and $T_2 = T_6$ can be derived, and during the deceleration phase, the equations $T_9 = T_{11} = T_{13} = T_{15}$ and $T_{10} = T_{14}$ exist. Since the velocity profile of the trajectory is symmetrical about the time variations, the time intervals of the acceleration and deceleration phases are equal; therefore, the motion profiles of the trajectory depend on four time intervals, i.e., the time intervals with a varying jerk $T_s$, constant jerk $T_j$, constant acceleration $T_a$, and constant velocity $T_v$. Therefore, the total execution time for the tracking of the overall trajectory is the summation of the 15 time intervals as follows:

$$T_f = \sum_{i=1}^{15} T_i = 8T_s + 4T_j + 2T_a + T_v \tag{4}$$

Let us define the rate of change in the jerk over time as a snap, with the maximum allowable snap being $s_{\max}$. The four previously mentioned key time intervals can be calculated with the following equations:

$$\begin{cases} T_s = \frac{j_{\max}}{s_{\max}} \\ T_j = \frac{a_{\max}}{j_{\max}} - T_s \\ T_a = \frac{v_{\max}}{a_{\max}} - T_j - 2T_s \\ T_v = \frac{q_s - q_e}{v_{\max}} - T_a - 2T_j - 4T_s \end{cases} \tag{5}$$

### 2.2. Calculation of the Time Parameter

When planning an S-curve trajectory with prescribed constraints, there are many different algorithms for optimizing the execution time. With a fifteen-segment trajectory, the time interval can be calculated by means of Equation (5); particular situations with different motion parameters and trajectory shapes have been reported [22,23]. For instance, a trajectory can be classified into different motion profiles to solve the numerical solutions of the motion parameters [22], and the expected solution will be selected from numerous cases, leading to complex solutions due to the multiple complicated symbolic equations. As shown in Figure 4, eight different profiles of the jerk of a four-order S-curve trajectory exist, which increases the computational burden in the selection of the appropriate trajectory with the aforementioned algorithm. Moreover, the time parameter can also be solved with an iterative algorithm with respect to the classified cases [23], as this can optimize the algorithm to reduce the computation cost; on the other hand, the iterative procedure will increase the time consumption, thus preventing real-time motion control.

In this work, a four-step algorithm is proposed to calculate $T_s$, $T_j$, $T_a$, and $T_v$; these time parameters are defined in Equation (5). According to the results that are calculated in each step, the trajectory type can be determined, and for different trajectory types, only the time interval of the necessary trajectory segment is essential for calculation in order to avoid redundant data processing for reduced time consumption. A flowchart of the algorithm is depicted in Figure 5 with a detailed interpretation below.

In the first step, by means of Equation (5), the maximum motion parameters can be determined if the time parameters $T_s$, $T_j$, $T_a$, and $T_v$ exist:

$$\begin{cases} j_{\max} = s_{\max} T_s \\ a_{\max} = s_{\max} T_s (T_s + T_j) \\ v_{\max} = s_{\max} T_s (T_s + T_j)(2T_s + T_j + T_a) \\ d_{\max} = s_{\max} T_s (T_s + T_j)(2T_s + T_j + T_a)(4T_s + 2T_j + T_a + T_v) \end{cases} \tag{6}$$

where $s_{\max}$ represents the maximum allowable snap, which is the second derivative of the acceleration [24], to evaluate the continuity of the jerk.

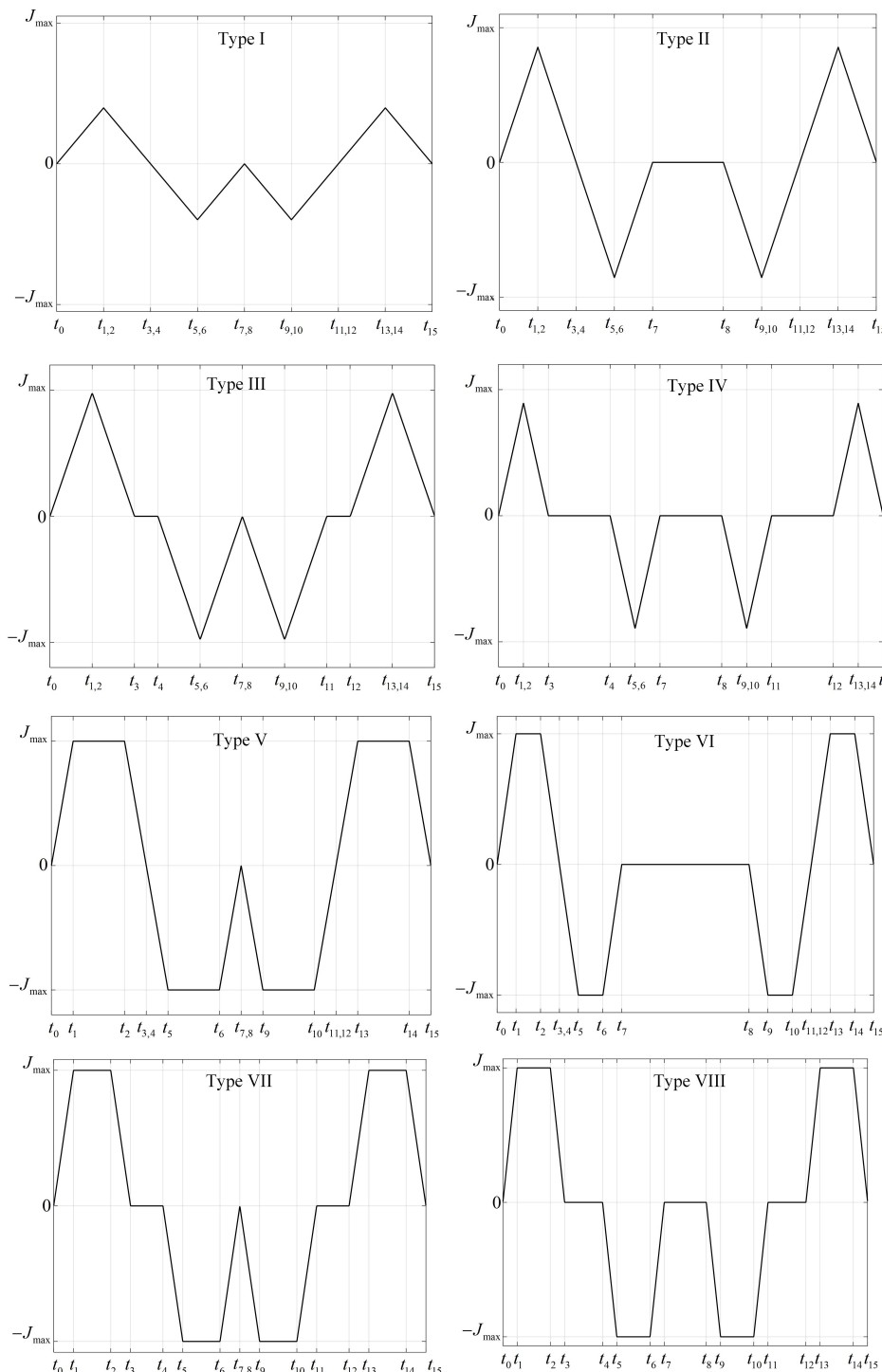

**Figure 4.** Different motion profiles when planning an S-curve trajectory.

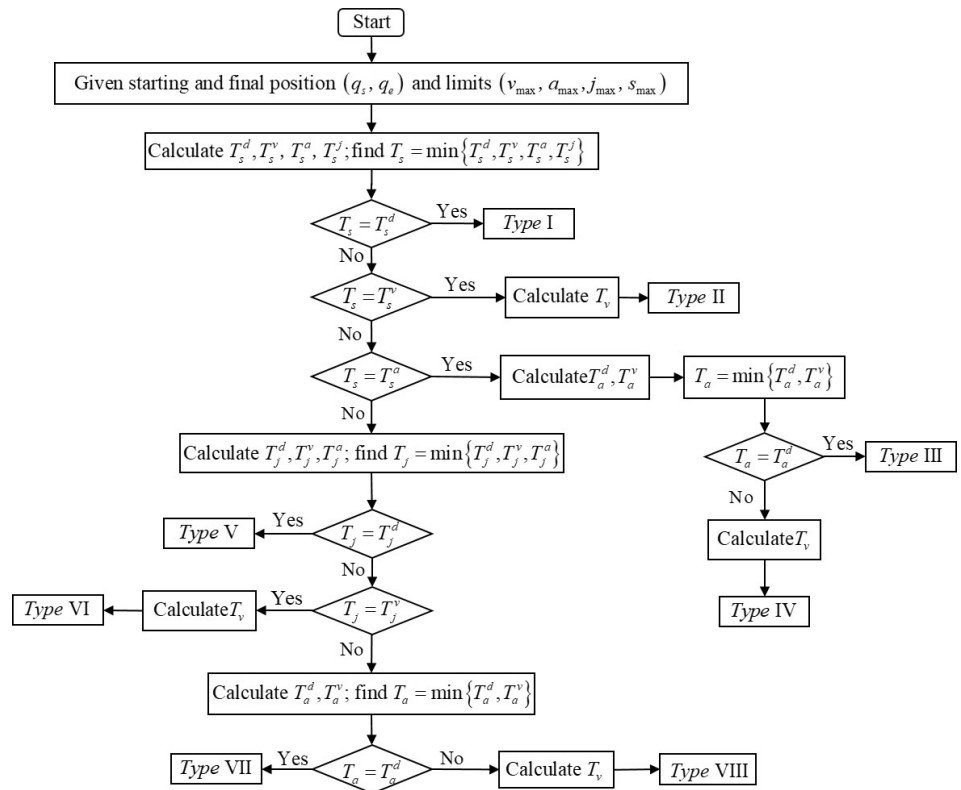

**Figure 5.** Flowchart of the proposed S-curve trajectory planning algorithm.

In the following steps, the calculation of the time intervals with different motion profiles will be presented.

### 2.2.1. Calculation of Time Interval $T_s$ with Varying Jerk

***Step 1***: Without considering the jerk, acceleration, and velocity constraints (i.e., $T_j$, $T_a$, and $T_v$ are nonexistent), the relationship between the maximum displacement $d_{max}$ and the snap $s_{max}$ can be derived based on Equation (6):

$$|d_{\max}| = |d(t_{15}|T_j = T_a = T_v = 0)| = 8T_s^4 s_{\max} \tag{7}$$

Moreover, the maximum displacement is calculated as $d_{max} = q_s - q_e$; then, the maximum time interval $T_s$ is calculated by using the displacement constraint:

$$T_s^d = \sqrt[4]{\frac{q_s - q_e}{8s_{\max}}} \tag{8}$$

***Step 2***: Similarly, with Equation (6), the relationship between the maximum velocity $v_{max}$ and the snap $s_{max}$ without the consideration of the jerk and acceleration constraints (i.e., $T_j$ and $T_a$ are nonexistent) can be solved as follows:

$$|v_{\max}| = |v(t_7|T_j = T_a = 0)| = 2T_s^3 s_{\max} \tag{9}$$

Thus, the boundary value of $T_s$ is calculated from the following velocity constraint:

$$T_s^v = \sqrt[3]{\frac{v_{\max}}{2s_{\max}}} \tag{10}$$

*Step 3*: By the same token, without considering the jerk constraint (i.e., $T_j$ is nonexistent), the maximum acceleration $a_{max}$ can be expressed in terms of $s_{max}$:

$$|a_{max}| = |a(t_3|T_j = 0)| = T_s^2 s_{max} \qquad (11)$$

The boundary value of $T_s$ is subject to the following acceleration constraint:

$$T_s^a = \sqrt{\frac{a_{max}}{s_{max}}} \qquad (12)$$

*Step 4*: Finally, the boundary value of $T_s$ is directly calculated with the jerk constraint:

$$T_s^j = \sqrt{\frac{j_{max}}{s_{max}}} \qquad (13)$$

To this end, the time interval of the jerk can be determined from the following minimum:

$$T_s = \min\{T_s^d, T_s^v, T_s^a, T_s^j\} \qquad (14)$$

In accordance with Equation (14), the time interval of the jerk may have the following four different solutions:

**Case 1:** If $T_s = T_s^d$, the displacement constraint is the only influencing factor that limits the motion time; thus, it is unnecessary to calculate the time intervals of the other trajectory segments, and there are only trajectory segments with varying jerks. In this case, the motion parameters, such as the jerk, acceleration, and velocity, cannot reach their maximum, and the real maximum jerk, acceleration, and velocity are equal to $j_m = s_{max} T_s^d$, $a_m = j_m T_s^d$, and $v_m = 2a_m T_s^d$, respectively.

**Case 2:** If $T_s = T_s^v$, the maximum velocity can be reached without the maximum jerk and acceleration. In this case, the maximal reachable jerk and acceleration are $j_m = s_{max} T_s^v$ and $a_m = j_m T_s^v$, respectively. For the calculation of the remaining motion parameters, the reader can refer to Section 2.2.4.

**Case 3:** If $T_s = T_s^a$, the maximum acceleration reaches its maximum without the maximum jerk, and the maximum reachable jerk is $j_m = s_{max} T_s^v$. For the calculation of the remaining motion parameters, the reader can refer to Section 2.2.3.

**Case 4:** If $T_s = T_s^j$, for the calculation of the remaining motion parameters, the reader can refer to Section 2.2.2.

### 2.2.2. Calculation of Time Interval $T_j$ with Constant Jerk

If Case 4 in Section 2.2.1 exists, i.e., $T_s = T_s^j$, and the maximum displacement $d_{max}$ can be written as a function of $T_j$ without considering the acceleration and velocity constraints from Equation (6), then:

$$|d_{max}| = |q_e - q_s| = |d(t_{15}|T_a = T_v = 0)| = j_{max}(8T_s^3 + 16T_s^2 T_j + 10T_s T_j^2 + 2T_j^3) \qquad (15)$$

Accordingly, the maximum jerk $j_{max}$ can be adopted to represent the displacement, since the time calculation is carried out to obtain the maximum allowable jerk, namely, $j_m = j_{max}$. The following calculations can be realized by using a similar approach.

With the displacement constraint, the boundary value of $T_j$ is calculated as

$$
|T_j^d| = \sqrt[3]{\frac{T_s^3}{27} + \frac{|q_e - q_s|}{4 j_{\max}} + \sqrt{\frac{|q_e - q_s| T_s^3}{54 j_{\max}} + \frac{|q_e - q_s|^2}{16 j_{\max}^2}}}
$$
$$
+ \sqrt[3]{\frac{T_s^3}{27} + \frac{|q_e - q_s|}{4 j_{\max}} - \sqrt{\frac{|q_e - q_s| T_s^3}{54 j_{\max}} + \frac{|q_e - q_s|^2}{16 j_{\max}^2}}} - \frac{5 T_s}{3}
\tag{16}
$$

The maximum velocity $v_{\max}$ can be calculated with $T_j$ without considering the acceleration constraint:

$$
|v_{\max}| = |v(t_7 | T_a = 0)| = j_{\max}(2 T_s^2 + 3 T_s T_j + T_j^2)
\tag{17}
$$

with the boundary value of $T_j^v$ being calculated according to the velocity constraint:

$$
T_j^v = \sqrt{\frac{T_s^2}{4} + \frac{v_{\max}}{j_{\max}}} - \frac{3 T_s}{2}
\tag{18}
$$

Moreover, the maximum acceleration is given by

$$
|a_{\max}| = a(t_3| = j_{\max}(T_s + T_j))
\tag{19}
$$

with the boundary value of $T_j$ from the acceleration constraint being

$$
T_j^a = \frac{a_{\max}}{j_{\max}} - T_s
\tag{20}
$$

Finally, the time interval $T_j$ for a motion with constant jerk is calculated as

$$
T_j = \min\{T_j^d, T_j^v, T_j^a\}
\tag{21}
$$

Similarly to Section 2.2.1, the selection of $T_j$ depends on the following three cases:

**Case 1:** If $T_j = T_j^d$, i.e., $T_a = T_v = 0$, only the trajectory segments with varying jerk and constant jerk exist, and the jerk can reach its maximum $j_{\max}$; therefore, the real maximum acceleration and velocity are calculated as $a_m = j_{\max}(T_s + T_j^d)$ and $v_m = a_m(2 T_s + T_j)$, respectively.

**Case 2:** If $T_j = T_j^v$, the maximum velocity can reach its maximum without the maximum acceleration, and the real maximum acceleration is calculated as $a_m = j_{\max}(T_s + T_j^v)$. For the calculation of the other motion parameters, the reader can refer to Section 2.2.4.

**Case 3:** If $T_j = T_j^a$, for the calculation of the time intervals, the reader can refer to Section 2.2.3.

2.2.3. Calculation of Time Interval $T_a$ with Constant Acceleration

As shown in Sections 2.2.1 and 2.2.2, the time intervals $T_s = T_s^j$ and $T_j = T_j^a$ exist. Without considering the velocity constraint, the maximum displacement $d_{\max}$ is calculated in terms of $T_a$ as

$$
|d_{\max}| = |d(t_{15} | T_v = 0)| = a_{\max}(8 T_a^2 + 3 T_a T_j + 6 T_s T_a + 8 T_s^2 + 2 T_j^2 + 8 T_s T_j)
\tag{22}
$$

Therefore, according to the displacement constraint, the boundary value of $T_a$ is solved as

$$
T_a^d = \frac{3 T_j}{2} - 3 T_s + \sqrt{\frac{(2 T_s + T_j)^2}{4} + \frac{q_e - q_s}{a_{\max}}}
\tag{23}
$$

Moreover, the maximum velocity is calculated as

$$|v_{\max}| = |v(t_7)| = a_{\max}(T_a + 2T_s + T_j) \tag{24}$$

and based on the velocity constraint, the boundary value of $T_a$ is solved as

$$T_a^v = \frac{v_{\max}}{a_{\max}} - T_j - 2T_s \tag{25}$$

Finally, the time interval $T_a$ with a constant acceleration is selected according to

$$T_a = \min\{T_a^d, T_a^v\} \tag{26}$$

In this case, the selection of $T_a$ depends on the following two cases:

**Case 1:** If $T_a = T_a^d$, i.e., $T_v = 0$, due to the limitation on displacement, trajectory segments with a constant velocity do not exist. Although the acceleration can reach its maximum, the velocity cannot, and the real maximum velocity is $v_m = a_{\max}(2T_s + T_a^d)$.

**Case 2:** If $T_a = T_a^v$, both the maximum acceleration and velocity can reach their maximums, and for the calculation of the motion parameters, the reader can refer to Section 2.2.4.

2.2.4. Calculation of Time Interval $T_v$ with Constant Velocity

Combining the previously calculated time intervals (e.g., $T_s$, $T_j$, and $T_a$) and Equation (26) leads to the time interval of the constant velocity:

$$T_v = \frac{q_e - q_s}{v_{\max}} - (4T_s + 2T_j + T_a) \tag{27}$$

So far, the four time intervals ($T_s$, $T_j$, $T_a$, and $T_v$) of the S-curve trajectory can be solved analytically case by case. In general, the kinematic constraints on robot joints are prescribed in real industrial applications; therefore, time intervals that are constrained by the acceleration and velocity can be calculated in advance and saved in the robot controller, wherein only the displacement constraints will be taken into consideration in real-time trajectory planning. Since the previously presented algorithm involves all of the trajectory profiles to meet the requirements of the kinematic constraints, it can accomplish trajectory planning with reliable and effective trajectories that are subject to the prescribed displacement. Moreover, the algorithm adopts the continuous trajectory curve of the jerk to ensure the smoothness of the curves of the acceleration and velocity; thus, it can avoid possible joint buffetings when the robot moves. In summary, the proposed algorithm can make the robot reach its maximum acceleration and velocity in the shortest time while aiming at a constrained time-optimal trajectory.

*2.3. Time Synchronization of Multi-Axis Motions*

As multi-dof mechanical systems, industrial robots are usually composed of multiple joints connected in series or parallel. In robot motion control, attention should be paid to the motion synchronization of each joint during the design of trajectories in the joint space to ensure the motion accuracy of the robot's end-effector, although each joint can be controlled independently. The planning algorithm presented in Section 2.2 can generate different time-optimal trajectories, but the execution time of each joint will be different due to the different kinematic constraints, which may result in some joints staying in motion while others stop moving, resulting in decreased robot compliance and trajectory tracking precision.

In light of the motion accuracy, it is essential to consider the time synchronization of the motions among the robot's joints during trajectory planning. Here, synchronization with respect to time means that all joints should complete their motions simultaneously when the robot executes its task, which can improve the stability of the robot when in

motion, in addition to ensuring trajectory tracking accuracy. Algorithm 1 depicts the corresponding trajectory planning algorithm for synchronization.

---

**Algorithm 1** Trajectory planning algorithm for multi-axis synchronization

---

**Input:** The initial $\mathbf{q}_s(t_0)$ and final $\mathbf{q}_e(t_f)$ positions of the robot and the kinematic constraints $\{\mathbf{v}_{max}, \mathbf{a}_{max}, \mathbf{j}_{max}\}$
**Output:** Time-synchronized motion trajectory

1: Use of the single-joint trajectory planning algorithm to calculate the execution time of trajectory planning for each joint $T_{f,k}$
2: Selection of the maximum execution time in step 1 as the synchronization time $T_f^{sync} = \max\{T_{f,1}, T_{f,2}, \ldots, T_{f,n}\}$
3: Calculation of the synchronization factor $\lambda_k = \frac{T_f^{sync}}{T_{f,k}}, k = 1, 2, \ldots, n$
4: Modification of the kinematic constraints of each joint $\{\mathbf{v}'_{max} = \frac{\mathbf{v}_{max}}{\lambda_k}, \mathbf{a}'_{max} = \frac{\mathbf{a}_{max}}{\lambda_k^2}, \mathbf{j}'_{max} = \frac{\mathbf{j}_{max}}{\lambda_k^3}\}$
5: Calculation of the new motion trajectory according to the new kinematic constraint $\{\mathbf{v}'_{max}, \mathbf{a}'_{max}, \mathbf{j}'_{max}\}$

---

In Algorithm 1, the first stage is the calculation of the minimum execution time of each joint according to the prescribed kinematic constraints; then, the largest one is selected as the normalized time to synchronize the motion trajectories of the remaining joints in order to make all of the joints adapt to the slowest joint's motion. In this procedure, the optimality of the execution time will be ignored; there are many methods for ensuring synchronous joint motion subject to kinematic constraints. The common approach to reducing the maximum motion parameters is to adopt an iterative algorithm to calculate the time intervals, but this has a high computational burden. In this work, a time-scaling method is applied to modify the geometric trajectory with the aim of handling the problem of parameter tuning, whereby the maximum velocity, acceleration, and jerk of a joint can be reached for an $n$-axis robot, while the remaining $n - 1$ robot joints will move within the same time. The time-synchronization algorithm is illustrated with a linear trajectory in the Cartesian space; the trajectory ends and the maximum motion parameters are listed in Table 2, and the planning results are shown in Figure 6.

**Table 2.** Linear trajectory and the maximum motion parameters.

| Axis | Initial Position [m] | Final Position [m] | Velocity [m/s] | Acceleration [m/s$^2$] | Jerk [m/s$^3$] |
|---|---|---|---|---|---|
| $x$ | 0 | 3 | 2 | 4 | 40 |
| $y$ | 0 | 2 | 1.4 | 5 | 35 |
| $z$ | 0 | 1.5 | 1.2 | 3 | 35 |

In Figure 6, it can be clearly seen that for the time-unsynchronized trajectory, the motion along the z-axis lasted for the longest time, and the motion along the x-axis took the shortest time. It is noted that the motion parameters along the three axes reached the constrained maximum values. On the other hand, the time-synchronized trajectory ensured that the same time was taken for motion along the three axes; the algorithm presented in Section 2.3 was adopted to adjust the original motions along the y- and z-axes to make the motion times of the latter two consistent with that of the x-axis. Moreover, the maximum motion parameters along the y- and z-axes (i.e., velocity, acceleration, and jerk) decreased along the synchronized trajectory compared to the original motion profiles.

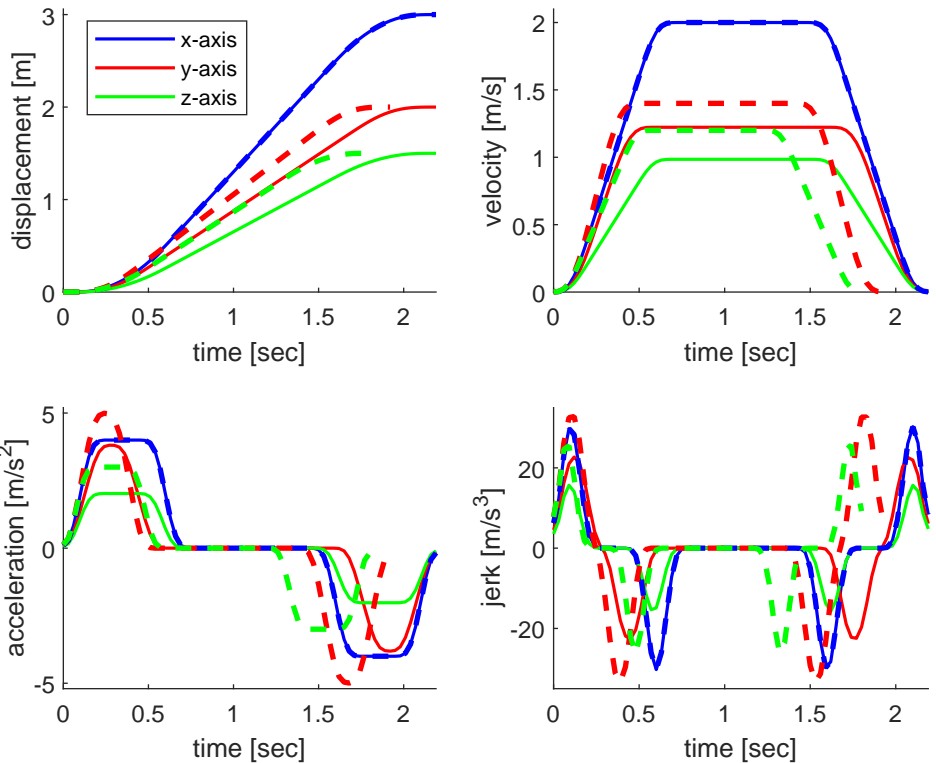

**Figure 6.** Comparison of the planned motion parameters along with a linear trajectory (solid lines—synchronized motion; dashed lines—unsynchronized motion).

## 3. Case Study: Trajectory Planning for a Five-Axis Manipulator

In this section, trajectory planning for a five-dof robotic manipulator that is to accomplish a PTP task is illustrated, as depicted in Figure 7. The trajectory planning was carried out in the joint space, where the angular positions of the joints were determined by solving the inverse kinematics according to the initial and final positions of the end-effector in the Cartesian space. Table 3 lists the boundary conditions of the task's requirements in the joint space and the kinematic limitations of the robotic joints.

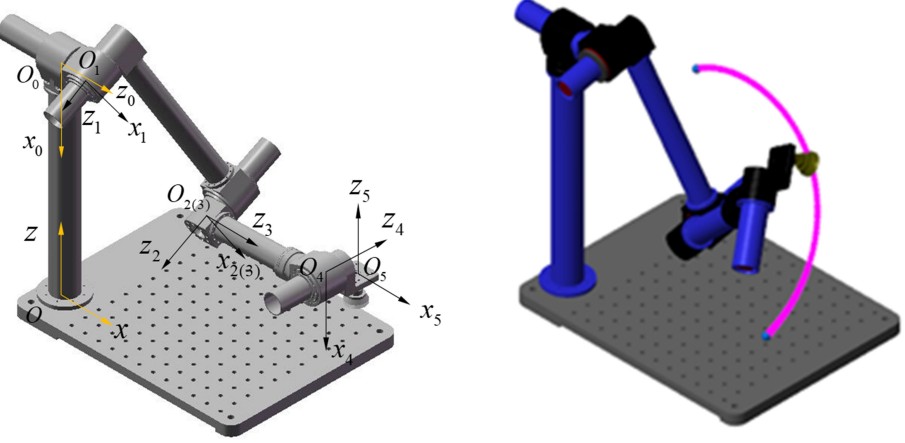

**Figure 7.** Coordinate systems of a five-dof manipulator and trajectory planning for a PTP task.

**Table 3.** Desired positions and kinematic motion limits of the robot's joints.

| | Joint | 1 | 2 | 3 | 4 | 5 |
|---|---|---|---|---|---|---|
| **Angular position** [rad] | **Initial** | 0 | 0 | 0 | 0 | 0 |
| | **Final** | $\pi/6$ | $\pi/4$ | $\pi/3$ | $\pi/2$ | $\pi/3$ |
| **Kinematic constraints** | **Velocity** [rad/s] | 1 | 1.4 | 1.4 | 2 | 3 |
| | **Acceleration** [rad/s$^2$] | 3 | 5 | 5 | 7 | 8 |
| | **Jerk** [rad/s$^3$] | 25 | 35 | 40 | 40 | 40 |

In the trajectory planning, the third-order and four-order S-curve trajectories were simulated in combination with the algorithm for multi-joint synchronized motions, and the results are shown in Figure 8. With the time-synchronized trajectory planning algorithms, the displacement, velocity, and acceleration profiles of all of the robotic joints for both trajectories were continuous, and the maximum motion parameters met the requirements of all of the prescribed kinematic constraints of the joints, which could ensure the continuity and stability of the robot's motion.

On the other hand, it was noteworthy that there was a sudden change in the acceleration curve of the third-order S-curve trajectory, and the jerks did not start and end with zero values, thus introducing a shock when the robot started or stopped moving. By contrast, the algorithm presented in this work was able to produce a trajectory with acceleration smoothness and jerk continuity, which could reduce these disadvantageous effects on the robot's motion. Moreover, the smoothness of the fourth-order S-curve trajectory was able to effectively avoid the problems of the impact and wear of the mechanical system caused by the frequent and rapid starting and stopping motions of the robot in the picking/placing applications.

To show the effectiveness of the proposed algorithm in S-curve trajectory planning, the computation times are listed in Table 4 and compared to those of different algorithms [25,26]. The computation time for the fourth-order S-curve in the case study was equal to 1.3461 s, which was lower than that of the quintic polynomial and cubic splines. The reason lies in that the algorithm was able to cause the robot to have a maximal driving performance for each joint while meeting its kinematic constraints. Moreover, the computation time for trajectory generation was close to that of the third-order S-curve trajectory. The maximum jerk appeared in the fourth joint, and the maximum jerk for the fourth-order S-curve was smaller than those in previously published works, but was slightly larger than that of the third-order one. It should also be noted that the generated trajectory was continuous on the $C_3$ level (i.e., the continuity level indicates the continuity of the motion parameters; for example, $C_2$ indicates acceleration continuity and $C_3$ indicates jerk continuity), rather than the $C_2$ level of the reported works [25–30]; thus, it could effectively avoid joint buffetings while programming the robot and could ensure the compliance of the robot with start–stop moments.

**Table 4.** Comparison of the results of multiple trajectory planning methods.

| Trajectory Model | Calculation Time [s] | Maximal Jerk [rad/s$^3$] | Continuity Level |
|---|---|---|---|
| Quintic polynomial [25] | 1.4667 | 40 (Joint 4) | $C_2$ |
| Cubic spline [26] | 2.1679 | 39.83 (Joint 4) | $C_2$ |
| Third-order S-curve | 1.2734 | 39.06 (Joint 4) | $C_2$ |
| Fourth-order S-curve | 1.3461 | 39.38 (Joint 4) | $C_3$ |

In terms of computational efficiency, the mean computation time for 100 trajectory generations for the five-dof robot was equal to 1.5 ms in the MATLAB environment, where the PC setup involved an Intel Core i5-9400F 2.90 GHz processor and 8 GB of memory. In light of the computational setup, the algorithm presented in this work can provide real-time motion control for robots.

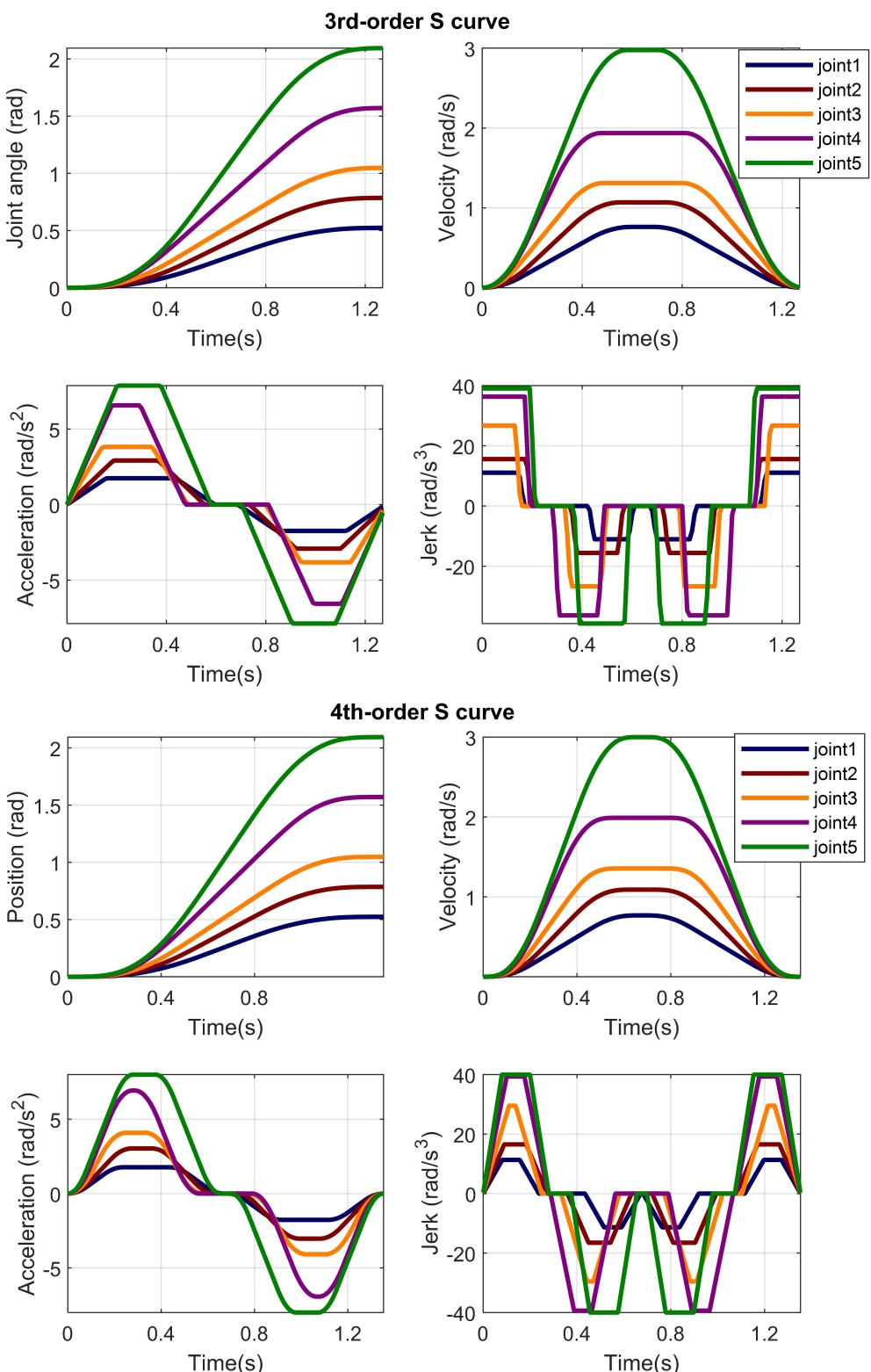

**Figure 8.** Comparison of planned trajectory for the five-DOF manipulator in terms of the motion profiles in the joint space.

## 4. Conclusions

In this work, a trajectory planning algorithm for robotic manipulators was proposed; it considered the jerk continuity, and the fourth-order S-curve was adopted to generate the trajectory to ensure smooth motion and to avoid the unexpected joint buffetings at

instantaneous moments when the robot would start to move. With its guaranteed trajectory flexibility, this algorithm can cause the acceleration and jerk to stay in a saturated state, which can improve the efficiency of robot programming. Moreover, a multi-axis synchronization planning algorithm was proposed and integrated for enhanced motion stability of a robot in terms of the generated synchronized and continuous motion trajectories. The effectiveness of the proposed trajectory planning algorithm was verified in both the joint and Cartesian spaces.

The algorithm proposed in this work does not involve any optimization procedures or iterative processes, as the kinematically constrained trajectory is generated by polynomial equations; thus, it features the advantages of real-time motion control for robots. The major advantage of the proposed algorithm lies in the generation of a smooth motion trajectory, which is subject to the optimal execution time, without the use of an iterative optimization algorithm. In addition, it can divide the whole trajectory into multiple smaller segments to speed up the acceleration/deceleration procedure in order to make full use of the motion performance to produce a superior trajectory. Moreover, the generated trajectory can allow the continuity level of jerk continuity to be realized, rather than the acceleration continuity reported in most other works; thus, it can effectively avoid shocks during the robot's movements. Future work will be devoted to extending the trajectory planning with the consideration of arbitrary boundary conditions that are suitable for applications that take spatially multi-segmented trajectories into account.

**Author Contributions:** Conceptualization, methodology, data curation, writing—review and editing, supervision, G.W.; software, validation, formal analysis, writing—original draft preparation, N.Z. All authors have read and agreed to the published version of the manuscript.

**Funding:** This research was funded by the Natural Science Foundation of Liaoning Province (grant number: 20180520028).

**Data Availability Statement:** Data available from the authors upon request.

**Conflicts of Interest:** The authors declare no conflict of interest.

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
