# Peer review of "Kinematically Constrained Jerk–Continuous S-Curve Trajectory Planning in Joint Space for Industrial Robots"

_electronics, doi:10.3390/electronics12051135_

Round 1

Reviewer 1 Report

the quality of the paper can be improved further by including following suggestions:

1. Real world implementation is must.

2. Newer Intelligent optimization algorithms have to be used.

3. Results have to be compared with literature.

4. More no. of case studies have to be considered.

5. Results and Discussion, Conclusion sections have to be elaborated.

Reviewer 2 Report

1. In the work, the authors proposed an algorithm for planning the S-curve trajectory for a manipulator with five-degree-of-freedom.
Is it possible to use too vthis algorithm to a manipulator with four-degree-of-freedom (four-DOF) or six-degree-of-freedom (six-DOF)?
Is the proposed algorithm for manipulators with degrees-of-freedom other than 5 (five-DOF) requires the use of changes for the profiles of  motion of S-curve trajectory planning adopted in it?

2. Have the authors in the paper considered the use of a kinematic S-curve trajectory of a higher order than 4?
Is it possible to use the same number of eight different profiles of the jerk in it as for a four-order S- curve trajectory?

3. In this work there is no comparison  of the proposed S-curve trajectory planning algorithm with algorithms involving an optimization procedure or an iterative process.
Please justify the advantages and profits of using the proposed algorithm in comparison to the previously mentioned algorithms.

4. On presented in the work figure 8, it can be seen that for the presented courses for planning the trajectory of the manipulator with five-degree-of-freedom (five-DOF) in terms of  the motion profiles in the joint space and for the S-curve of the 3rd order as well as 4th order, no changes in the amplitude values ​​are visible, but only changes in their shapes and only for acceleration and jerk.

5. In which cases algorithms involving optimization procedures or an iterative process be used more should than the proposed S-curve trajectory planning algorithm?

6. Are there other advantages of using polynomial equations to generate a kinematically constrained trajectory for the proposed trajectory planning algorithm for robotic manipulators, apart from real-time motion control of robots. If so, please write which ones.

Reviewer 3 Report

The article presents a robot trajectory planning algorithm using a polynomial. The structure of the article is appropriate. I have the following comments 

Line 34 what does 4-4-7-4 mean? explain in the text 

Equation 4 explain what s_max, snap is not a popular parameter explanation is advisable 

Line 121 - 135 fix the formula for jerk I think there should be a snap used there jm=s_max *T ? 

Table 1  fix unit for jerk 

Figure 3 and  table 2 How to understand joints O2-3 in the figure is one in the table there are two columns please explain

Line 239 and Table 3. How Continuity level is defined explain 

Table 3 When comparing, you specify the calculation time. How was this time determined ? I understand that your algorithm was simulated, but the algorithm from the classic article 24 was also implemented on a real robot. Was your algorithm implemented on a robot - such a test would significantly increase the scientific quality of your article 

In the conclusion and in the abstract, you should provide quantitative values that distinguish your algorithm 

END

Round 2

Reviewer 1 Report

congratulations